# LncRNA *TCONS_00323213* Promotes Myogenic Differentiation by Interacting with PKNOX2 to Upregulate *MyoG* in Porcine Satellite Cells

**DOI:** 10.3390/ijms24076773

**Published:** 2023-04-05

**Authors:** Mengxun Li, Quan Liu, Su Xie, Chong Fu, Jiaxuan Li, Cheng Tian, Xin Li, Changchun Li

**Affiliations:** 1Key Laboratory of Agricultural Animal Genetics, Breeding and Reproduction of the Ministry of Education, Huazhong Agricultural University, Wuhan 430070, China; 2Key Laboratory of Swine Genetics and Breeding of the Ministry of Agriculture, Huazhong Agricultural University, Wuhan 430070, China; 3The Cooperative Innovation Center for Sustainable Pig Production of Hubei Province, Huazhong Agricultural University, Wuhan 430070, China

**Keywords:** *TCONS_00323213*, PKNOX2, porcine skeletal muscle satellite cells, differentiation, myogenesis

## Abstract

Myogenic differentiation is a complex biological process that is regulated by multiple factors, among which long noncoding RNAs (lncRNAs) play an essential role. However, in-depth studies on the regulatory mechanisms of long noncoding RNAs (lncRNAs) in myogenic differentiation are limited. In this study, we characterized the role of the novel lncRNA *TCONS_00323213*, which is upregulated during porcine skeletal muscle satellite cell (PSC) differentiation in myogenesis. We found that *TCONS_00323213* affected the proliferation and differentiation of PSC in vitro. We performed quantitative polymerase chain reaction (qPCR), 5-ethynyl-20-deoxyuridine (EdU), western blotting, immunofluorescence staining, pull-down assays, and cleavage under targets and tagmentation (CUT and Tag) assays to clarify the effects and action mechanisms of *TCONS_00323213*. LncRNA *TCONS_00323213* inhibited myoblast proliferation based on analyses of cell survival rates during PSC proliferation. Functional analyses revealed that *TCONS_00323213* promotes cell differentiation and enhances myogenin (*MyoG*), myosin heavy chain (*MyHC*), and myocyte enhancer factor 2 (*MEF2C*) during myoblast differentiation. As determined by pull-down and RNA immunoprecipitation (RIP) assays, the lncRNA *TCONS_00323213* interacted with PBX/Knotted Homeobox 2 (PKNOX2). CUT and Tag assays showed that PKNOX2 was significantly enriched on the *MyoG* promoter after lncRNA *TCONS_00323213* knockdown. Our findings demonstrate that the interaction between lncRNA *TCONS_00323213* and PKNOX2 relieves the inhibitory effect of PKNOX2 on the *MyoG* promoter, increases its expression, and promotes PSC differentiation. This novel role of lncRNA *TCONS_00323213* sheds light on the molecular mechanisms by which lncRNAs regulate porcine myogenesis.

## 1. Introduction

Skeletal muscle growth and development are regulated by a variety of genes and epigenetic factors [1]. Porcine skeletal muscle satellite cells (PSC) are the fundamental unit for skeletal muscle formation [2,3,4]. The growth, development, and regeneration of skeletal muscle depends on the activation state of skeletal muscle satellite cells, which proliferate and differentiate for repair when skeletal muscle is stimulated by exercise or injury and exhibits self-renewal [3,5,6]. Owing to these crucial roles in the growth, development, and regeneration of skeletal muscle, satellite cells are an ideal material to study myogenic differentiation.

Long noncoding RNAs (lncRNAs) are important regulators of skeletal muscle growth and development [7,8]. There is increasing evidence that lncRNAs participate in the regulation of skeletal muscle development, myoblast proliferation and differentiation, and related biological processes at transcriptional, post-transcriptional, translational, and epigenetic levels. *H19*, a well-known imprinted lncRNA [9], is expressed in animal skeletal muscle and heart tissues and exerts important functions [10]. *H19* can inhibit insulin-like growth factor 2 (*IGF2)* transcription by binding to polycomb repressive complex 2 (PRC2) [11] and recruit TAR DNA-binding protein 43 (TDP43) and drebrin 1 (DBN1) to induce differentiation of PSCs [12,13]. The lncRNA metastasis-associated lung adenocarcinoma transcript 1 (*Malat1)* functions via the suppressor of variation 39 homolog 1 (Suv39h1) to repress the expression of the target gene myogenic differentiation factor (*MyoD)* [14]. Developmental pluripotency-associated 2 (*Dppa2*) upstream binding muscle (*Dum)* is a lncRNA expressed in myogenic cells. During *MyoD*-induced differentiation of myogenic cells, *Dum* recruits large amounts of DNA methyltransferases (DNMTs) to the *Dppa2* promoter, thereby silencing *Dppa2* expression and promoting myogenesis [15]. Maternally expressed gene 3 (*Meg3)* is a homologous heterozygous lncRNA of *Gtl2* in humans and promotes PSCs by interacting with *miR-423-5p* to relieve the inhibitory effect on serum response factor (*SRF)* [16]. It was shown that synaptopodin-2 (*SYNPO2)* intron sense-overlapping lncRNA (*lncSYISL*) and muscle growth-promoting factor (*lncMGPF)* are directly regulated by *MyoD* [17,18]. *lncSYISL* can repress target gene expression and promote cell proliferation by recruiting PRC2, leading to H3K27 in the target gene promoter regions [17]. *lncMGPF* competes with myocyte enhancer factor 2 (*MEF2C)* to bind *miR-135a-5p* and promote myogenic differentiation. Furthermore, *lncMGPF* enhances the binding ability of human antigen R (HuR) proteins to myogenic regulatory mRNAs while enhancing the stability of these mRNAs, thereby promoting myoblast differentiation [18]. In skeletal muscle, the lncRNA taurine-upregulated gene 1 (*Tug1*) interacts with the peroxisome proliferator-activated receptor-γ co-activator-1α (PGC-1α) in the regulation of transcriptional responses to exercise [19]. The novel lncRNA *Gm10561* was confirmed to sponge *miR-432* to modulate *MEF2C* and E2F transcription factor 3 (*E2F3)* expression to regulate myoblast proliferation and differentiation [20]. These findings indicate the important contributions of lncRNAs to myogenesis and skeletal muscle development.

The lncRNA *TCONS_00323213* on chromosome 15 is an intergenic lncRNA [21]. A recent report used published RNA-Seq data to perform transcriptome analysis of PSCs differentiated 24 and 36 h after *TCONS_00323213* knockdown, the results showed that *TCONS_00323213* participates in the regulation of PSC differentiation by regulated gene expression and alternative splicing events [22]. Therefore, we speculate that *TCONS_00323213* is essential and required for PSC differentiation. In this study, we further explored the function and molecular mechanism of action of *TCONS_00323213* in myogenic PSCs. In particular, we conducted pull-down assays and mass spectrometry (MS) analysis to identify the protein that binds to *TCONS_00323213*, followed by additional analyses of the role of this binding protein in the myogenic differentiation of PSCs. The results improve our understanding of the functions of lncRNA *TCONS_00323213* and its regulatory axis in the differentiation of PSCs.

## 2. Results

### 2.1. TCONS_00323213 Inhibits Myoblast Proliferation

We knocked down and overexpressed *TCONS_00323213* in PSCs in the proliferation phase and performed EdU staining and CCK-8 assays. We used three phosphorothioate-modified antisense oligonucleotide (ASO) sequences (ASO-*3213*-1, ASO-*3213*-2, and ASO-*3213*-3) to knockdown *TCONS_00323213*. The most effective ASO sequence was ASO-*3213*-1 (Figure 1A) (Appendix A), and we used it for subsequent experiments. The ASO group had higher EdU incorporation than the control group (Figure 1C), suggesting that knocking down *TCONS_00323213* can increase mitotic activity. We cloned *TCONS_00323213* into the PZW1 plasmid and successfully constructed the overexpression vector PZW1-*TCONS_00323213* (Figure 1B). Overexpressing *TCONS_00323213* decreased EdU binding efficiency and mitotic activity (Figure 1D). A CCK-8 assay revealed that *TCONS_00323213* knockdown significantly accelerated cell proliferation (Figure 1E). In contrast, *TCONS_00323213* overexpression significantly diminished the proliferation of PSCs compared with that in the negative control group (Figure 1F).

### 2.2. TCONS_00323213 Promotes Myoblast Differentiation

To further investigate the effect of *TCONS_00323213* on the differentiation of PSCs. We have detected the expression of *TCONS_00323213* at the different tissue and different time points of proliferation and differentiation of PSCs. The result showed *TCONS_00323213* is specific expression in skeletal muscle and heart (Appendix A), and it increases gradually during myogenic differentiation (Appendix A). Additionally, qPCR results demonstrated that the mRNA expression levels of myogenic differentiation-related genes were significantly down regulated under *TCONS_00323213* knockdown (Appendix A). These results implied that *TCONS_00323213* plays an important role in myogenic differentiation. Then we detected the expression of MyoD, MyoG, and MEF2C at the protein level after knockdown and overexpression of *TCONS_00323213*. As determined by western blot, the protein expression levels of MyoG and MEF2C decreased significantly after *TCONS_00323213* knockdown (Figure 2A) and increased significantly after *TCONS_00323213* overexpression (Figure 2B). Immunofluorescence results demonstrated that MyoG expression was significantly downregulated after *TCONS_00323213* knockdown (Figure 2C) and significantly upregulated after *TCONS_00323213* overexpression compared with levels in the control group (Figure 2D).

### 2.3. TCONS_00323213 Interacts Directly with PBX/Knotted Homeobox 2

To explore the function of *TCONS_00323213*, an RNA pull-down assay was performed to identify the interacting proteins of *TCONS_00323213* (Figure 3A). Given that the *TCONS_00323213* full-length sequence is excessively long, we truncated *TCONS_00323213* into 5 biotinylated fragments (Figure 3B) and incubated these fragments with cell lysates from differentiated PSCs for 36 h. RNA-binding proteins were captured with streptavidin magnetic beads. Among the differentially expressed proteins detected by MS, the transcription factor PBX/Knotted Homeobox 2 (PKNOX2) was of particular interest. PKNOX2, a member of the PREP family, is highly expressed in skeletal muscle tissue [23]. The western blot showed that PKNOX2 was captured by biotinylated *TCONS_00323213* fragments 1, 2, and 3 (Figure 3C). The results indicated that PKNOX2 was precipitated by biotinylated *TCONS_00323213*. We then performed the RIP assay to confirm this interaction (Figure 3D). As expected, the PKNOX2 protein captured *TCONS_00323213* in the PSC lysate (Figure 3E,F), indicating that PKNOX2 interacts with *TCONS_00323213* in vivo. PKNOX2 is a member of the triple amino acid loop extension (TALE) superfamily of homology domains and is highly similar to its homologous protein PKNOX1 in terms of structural domains. It specifically recognizes the TGACAG motif [23]. We subsequently analyzed the sequence characteristics of *TCONS_00323213* and found 2 TGACAG motifs located at bases 44–49 and 4381–4386 in *TCONS_00323213* (Figure 3B), so we speculated that PKNOX2 can bind to *TCONS_00323213* at the TGACAG sequence.

We further performed fluorescence in situ hybridization (FISH) and IF to further investigate the regulatory mechanisms of *TCONS_00323213* and PKNOX2 in PSCs. The results showed that *TCONS_00323213* was mainly expressed in the nucleus and present at low levels in the cytoplasm, and PKNOX2 was also mainly localized in the nucleus (Figure 3G). These findings indicate that *TCONS_00323213* and PKNOX2 may interact mainly in the nucleus. In addition, we examined the expression of PKNOX2 in PSCs after the *TCONS_00323213* knockdown. The IF results showed that the expression of PKNOX2 in the cytoplasm of PSCs increased after *TCONS_00323213* knockdown, and the fluorescence expression intensity increased significantly as well (Figure 3H). This result suggests that *TCONS_00323213* may interact with PKNOX2 to affect the nuclear retention of PKNOX2 and thus participate in the regulation of PSC differentiation.

### 2.4. PKNOX2 Affects PSCs Differentiation

To explore the function of *PKNOX2* in PSCs, we detected the expression pattern of *PKNOX2* in PSCs at different proliferation and differentiation time points. The results showed that the *PKNOX2* expression level in the proliferation periods was higher than in the differentiation periods (Figure 4A). This result indicates that PKNOX2 functions in the differentiation of PSCs. Then, we designed and synthesized two siRNA sequences targeting *PKNOX2* (si-*PKNOX2*-1 and si-*PKNOX2*-2) (Appendix A). After transfection, si-*PKNOX2*-2 had the highest knockdown efficiency (Figure 4B). Therefore, si-*PKNOX2*-2 was used in all subsequent experiments for *PKNOX2* knockdown. We also constructed an effective overexpression vector for *PKNOX2* via homologous recombination (Figure 4D). Subsequently, skeletal muscle satellite cells were transfected with si-*PKNOX2*-2, and the overexpression vector PZW1-*PKNOX2* and the expression of related marker genes were examined after 36 h of PSC differentiation. The qPCR results confirmed that the mRNA expression levels of *TCONS_00323213* did not significantly change by knockdown and overexpression of PKNOX2 (Figure 4C,E). However, *MyoG* and *MEF2C* were significantly upregulated by PKNOX2 knockdown (Figure 4C) and significantly downregulated by overexpression of PKNOX2 (Figure 4E). The western blot illustrated that the knockdown of PKNOX2 significantly upregulated the protein expression levels of MyoG, MEF2C, and MyHC (Figure 4F,G). In contrast, overexpression of PKNOX2 significantly downregulated the protein expression levels of MyoG, MEF2C, and MyHC (Figure 4H,I). Immunofluorescence results demonstrated that MyHC expression was significantly upregulated after PKNOX2 knockdown (Figure 4J) and downregulated by PKNOX2 overexpression (Figure 4K). These results suggest that PKNOX2 knockdown promotes the differentiation of PSCs.

### 2.5. LncRNA TCONS_00323213 Decreased the Enrichment of PKNOX2 on the MyoG Promoter

Our experimental results demonstrated that *TCONS_00323213* interacts with PKNOX2. Next, we identified the downstream target sites of PKNOX2 after the knockdown of *TCONS_00323213* by CUT and Tag experiments. We transfected PSCs with ASO to knock down *TCONS_00323213* and conducted the CUT and Tag assay after 36 h of induced differentiation to obtain a genome-wide binding site map for PKNOX2 (Figure 5A). The genome-wide analysis of binding sites showed that PKNOX2 was mainly enriched in promoter regions (<2 kb, ASO-NC: 35.27%, ASO-*TCONS_00323213*: 33.80%) (Figure 5C). A motif analysis showed that PKNOX2 was significantly enriched in TGACAG motifs (Figure 5B), indicating that the *TCONS_00323213*–PKNOX2 complex participates in the regulation of gene expression.

To investigate how the *TCONS_0032321*3–PKNOX2 complex regulates the expression of genes, we first analyzed the RNA-seq of PSCs for 36 h of differentiation after knockdown of *TCONS_00323213* (accession number: SRP186451) and obtained 12,111 differentially expressed genes (Appendix A). Then, we compared PKNOX2’s enriched peaks in the control and *TCONS_00323213* knockdown groups and obtained 833 target genes (Appendix A) of PKNOX2 by a CUT and Tag analysis. A total of 307 significantly differentially expressed genes (Appendix A) were obtained by combining the target genes of PKNOX2 with differentially expressed genes identified by the RNA-seq of PSCs for 36 h of differentiation after knockdown of *TCONS_00323213* (Figure 5D). These genes were significantly enriched in KEGG pathways and biological processes related to skeletal muscle growth and regeneration, such as ECM–receptor interaction, focal adhesion, thyroid hormone, PI3K–Akt, MAPK, Hippo, and TGF-β signaling pathways (Figure 5E,F).

Combined with Integrative Genomics Viewer (IGV) analysis, we found that *MyoG* and *MYC*, which are myogenic-related genes, had significantly different PKNOX2 enrichment peaks in their promoter regions (Figure 6A,B). Therefore, we hypothesized that *TCONS_00323213* regulates gene expression through binding to PKNOX2, which in turn affects PSC differentiation. Given that enrichment of PKNOX2 peaks on the *MyoG* promoter increased significantly after the *TCONS_00323213* knockdown, a luciferase report assay was performed to further verify this relationship. We found that the overexpression of PKNOX2 reduced the luciferase activity of the wild-type *MyoG* promoter and that luciferase activity increased upon *TCONS_00323213* overexpression after PKNOX2 and the *MyoG* promoter plasmid had been co-transfected with the *MyoG* promoter (Figure 6C). Overexpression of PKNOX2 did not affect the luciferase activity of the *MyoG* promoter mutant (Figure 6D). Altogether, these results showed that the lncRNA *TCONS_00323213* interacted with PKNOX2 to relieve the inhibitory effect of PKNOX2 on *MyoG*, thereby promoting PSC differentiation (Figure 7).

## 3. Discussion

Skeletal muscle growth is a complex and precise process that is tightly linked to gene expression regulation. Numerous transcription factors, such as Pax7, Myf5, MyoD, MyoG, and MEF2C, are involved in the regulation of myogenesis [24,25,26]. Moreover, a growing number of lncRNAs and related binding proteins involved in this process have been identified [27,28,29]. In our study, we found *TCONS_00323213* to be specifically expressed in porcine skeletal muscle and cardiac tissues and regularly expressed in the differentiation period of PSCs. A recent report used public RNA-seq data and performed a LivestockExp transcriptome analysis of PSC differentiation at 24 and 36 h after the *TCONS_00323213* knockdown. This research reported 18,436 differentially expressed genes and 2036 differential alternative splicing events in PSC differentiation at 24 h and 14,635 differentially expressed genes and 2193 differential alternative splicing events in PSC differentiation at 36 h. This report implies that *TCONS_00323213* participates in the regulation of PSC differentiation through regulated gene expression and alternative splicing events [22]. In our study, we analyzed RNA-seq data only 36 h after *TCONS_00323213* knockdown, and we obtained 12,111 differentially expressed genes. The reason for the different number of differentially expressed genes obtained may be due to the different analysis methods and screening conditions. Our results also demonstrate that *TCONS_00323213* can be involved in satellite cell differentiation by regulating gene expression. In addition, our study explored the regulatory mechanism of *TCONS_00323213* on PSCs in more detail through a series of molecular biology experiments. This further indicates that *TCONS_00323213* plays a very important role in the regulation of myogenic cell differentiation.

PBX/knotted 1 homeobox 2 (PKNOX2) was first discovered in 2001 and lies on the chromosomal region 11q24 in humans. It is a nuclear transcription factor belonging to the TALE class of homologous domain proteins. PKNOX2, which is similar to PKNOX1, specifically recognizes the TGACAG motif and plays key roles in cell growth, differentiation, and apoptosis [23,30]. In this study, localization experiments of the PKNOX2 protein showed that it is mainly present in the nucleus, which is consistent with the results of previous studies. Research on PKNOX2 has focused on its roles in diseases, such as inhibition of tumor cell migration, proliferation by PKNOX2 promoter demethylation, and induction of apoptosis [31]. Studies have shown that PKNOX2 knockdown can accelerate the death of myofibroblasts and proximal tubular epithelial cells [32]. *LINC02489* interacts with PKNOX2 through the PTEN/mTOR axis to reduce the migration and invasion of chemotherapy-resistant SKOV3 cells, thereby increasing the sensitivity of ovarian cancer to paclitaxel [33]. However, there is a scarcity of research on the role of PKNOX2 in pig muscle development. In the present study, a series of experimental assays demonstrated that the PKNOX2 protein interacts with *TCONS_00323213* to affect PSC differentiation. This provides new insights into the regulation of cell differentiation by PKNOX2.

Recent studies revealed that lncRNAs have diverse functions in muscle development and myoblast-directed differentiation [34,35,36,37]. As with most noncoding RNAs with regulatory functions, the majority of lncRNAs exert functions in the nucleus by directing RNA complexes to specific RNA, DNA, or protein targets [38,39,40,41]. For example, *lncSYISL* can recruit the enhancer of the zeste homolog 2 (EZH2) protein from the PRC2 complex to the promoter regions of muscle development-related target genes, such as cyclin-dependent kinase inhibitor 1A (*p21*) and *MyoG*, thereby repressing the transcription of these genes and inhibiting cell differentiation [17]. LncRNA muscle atrophy-associated transcript (*lncMAAT*) negatively regulates transcription of *miR-29b* through the sex-determining region Y *(SRY)* related high-mobility group box 6 (*SOX6*) by a trans-regulatory module and increases the expression of the neighboring gene *Mbnl1* by a cis-regulatory module [42]. The lncRNA *OIP5* antisense RNA 1 (*OIP5-AS1)* is expressed in muscle tissues and acts as an interactive scaffold, recruiting RNA binding protein (RBP) HuR to bind to the *MEF2C* 3′ UTR region, stabilizing *MEF2C* mRNA, and promoting myogenic differentiation [43]. By influencing the transcriptional regulation of its neighboring gene, filamin A-interacting protein 1 (*Filip1),* and by specifically binding to the TDP-43 protein, muscle-enriched lncRNA *(lncMyolinc*) promotes the expression of α-actin (*Acta1)* and *MyoD* and thus contributes to muscle development [44]. *Myoparr* is a nuclear retention protein that acts as a protein scaffold, enhancing the binding of the DEAD-box 17/P300-CBP-associated factor (Ddx17/PCAF) protein to the *MyoG* promoter, which in turn activates *MyoG* expression and promotes myogenic differentiation [45]. LncRNA *Irm* regulates the expression of myogenic genes by directly binding to myocyte enhancer factor 2D (MEF2D), which in turn promotes the assembly of MyoD/MEF2D on the regulatory elements of target genes [27]. In our study, we demonstrate that the interaction between lncRNA *TCONS_00323213* and PKNOX2 relieves the inhibitory effect of PKNOX2 on the *MyoG* promoter and promotes PSC differentiation. This provides new evidence for how lncRNA and proteins interact to regulate myogenic differentiation.

In reviewing the results of this study, some potential limitations should be con-sidered. First, we more focused on the function of TCONS_00323213 on PSCs differen-tiation, but failed to explored deeply into the molecular mechanism of TCONS_00323213 in regulating PSCs proliferation. Second, we found TCONS_00323213 affects the locali-zation of PKNOX2 protein in PSCs, but specific mechanism has not to be clearly. In addition, the fact that TCONS_00323213 was also detected in the cytoplasm implies that it can regulate myogenesis though other mechanisms. All these limitations deserve further investigation in the future.

## 4. Materials and Methods

### 4.1. Ethics Statement

In this study, animal care and all experiments were carried out in accordance with the pre-approved guidelines of Regulation Proclamation No. 5 of the Standing Committee of the Hubei People’s Congress. All experimental protocols were approved by the Ethics Committee (HZAUSW2015-0003) of Huazhong Agricultural University, Wuhan City, Hubei Province, China.

### 4.2. PSCs Culture and Transfection

Fresh PSCs were isolated from the hind legs of male Yorkshire piglets less than 1 week of age. In accordance with the methods of Wang et al. [46], tissue block digestion was used to isolate skeletal muscle satellite cells in a sterile environment. Muscle tissue was cut into pieces and then digested with 320 U/mL collagenase type II (Gibco, LA, CA, USA) in a 37 °C water bath with shaking for 2 h. After termination with DMEM (Gibco, LA, CA, USA) that contained 10% FBS (Gibco, LA, CA, USA), the cell suspension was filtered through 400-, 200-, 100-, and 50-µm filters to remove tissue debris. The filtrate was retained. Afterwards, the resulting cell pellet was resuspended and cultured in PM+ containing an RPMI-1640 medium supplemented with 20% FBS, 0.5% GlutaMax, 0.5% nonessential amino acids, 0.5% Anti-Anti (Gibco, LA, CA, USA), 0.25% chicken embryo extract (Gemini, Woodland, CA, USA), and 2.5 ng/mL basic fibroblast growth factor (Invitrogen, Grand Island, NY, USA). When the PSCs grew to 90% confluence, they were transferred into DMEM supplemented with 5% horse serum (Gibco, LA, CA, USA) to induce differentiation. All cells described above were incubated at 37 °C under 5% CO_2_.

When cell confluence reached 80%, cells were transfected with 200 nM (final concentration) of ASO-negative control (*NC*)/ASO-*TCONS_00323213* (Genepharma, Suzhou, China), 100 nM of si-*NC*/si-*PKNOX2* (Genepharma, Suzhou, China), or 2.5 μg of plasmid per well (6-well culture plate) using jetPRIME (Polyplus, Illkirch, France) according to the user manual. Transfection efficiency was measured using a PZW1 vector expressing GFP or an NC after 36 h of transfection, and qPCR assays were performed.

### 4.3. EdU Assay

EdU, a thymidine nucleoside analog, can be used to detect cellular DNA replication activity during DNA replication in living cells through its specific reaction with Apollo^®^ fluorescent dyes. When the PSCs grew to approximately 80% density in 10-cm culture plates, they were transferred to 12-well plates. *TCONS_00323213* knockdown and overexpression were performed when the cell density reached 40–50%. The EdU (RiboBio, Guangzhou, China) reagent was added to each well at a final concentration of 50 μM for 1.5–2 h. Then, the cells were fixed with 4% paraformaldehyde solution and 0.5% Triton X-100 and subsequently incubated with apollo staining solution for 30 min at 25 °C and protected from light. Nuclei were lightproof-stained with 4′,6-diamidino-2-phenylindole (DAPI) for 10 min and finally examined and photographed under a Leica SP8 fluorescent microscope.

### 4.4. CCK-8 Assay

PSCs were seeded into 96-well plates at approximately 2500 cells per well with 100 µL of PM+. Then, cell proliferation was detected by the CCK-8 assay kit (Vazyme, Nanjing, China) in accordance with the manufacturer’s instructions. Absorbance at 450 nm was measured using a spectrophotometer after 0, 12, 24, 36, and 48 h of transfection.

### 4.5. RNA Extraction and RT-qPCR Analysis

Total RNA was extracted from cultured PSCs using a StayPure RNA Extraction Kit (Accurate, Changsha, China). The total RNA from the pull-down and RIP assays was isolated using TRIzol reagent (Vazyme, Nanjing, China) in accordance with the manufacturer’s protocol. For cDNA synthesis, RNA was reverse-transcribed using ABScript III RT Master Mix for qPCR with gDNA Remover (ABconal, Wuhan, China). qPCR for RNA was carried out with 2 × Universal SYBR Green Mix Fast qPCR Mix (ABconal, Wuhan, China) on a Bio-Rad CFX96 Real-Time Detection System, following the manufacturers’ instructions. The 2^−∆∆CT^ method was used to analyze qPCR data. The primers used for qPCR in this study were designed using Primer 5.0 (Appendix A).

### 4.6. Western Blot Analysis

The total protein of cells was lysed in a RIPA buffer containing 1% protease inhibitors (PMSF) and collected. After SDS-PAGE, the proteins were transferred onto polyvinylidene fluoride membranes. The primary antibodies were anti-PKNOX2 (Proteintech, Wuhan, China), anti-MyoG (Abcam, Cambridge, UK), anti-MEF2C (Proteintech, Wuhan, China), anti-MyHC (Millipore, Billerica, MA, USA), and anti-β-tubulin (Servicebio, Wuhan, China). The HRP-conjugated secondary antibodies HRP-labeled goat anti-mouse IgG (Servicebio, Wuhan, China) and HRP-labeled goat anti-rabbit IgG (Servicebio, Wuhan, China) were used.

### 4.7. Immunofluorescence Assay

PSCs were washed 2 times with phosphate-buffered saline (PBS) and fixed with ice-cold 4% paraformaldehyde for 15 min and subsequently incubated in 0.3% Triton X-100 at room temperature for 10 min. Then, the PSCs were blocked with a blocking solution (3% bovine serum albumin (BSA), 0.3% Triton X-100, and 10% FBS complemented with PBS) at 37 °C for 0.5 h and washed 2–3 times with PBS. Next, the PSCs were incubated with anti-MyoG (Abcam, Cambridge, MA, USA) and MyHC antibodies (Millipore, Billerica, MA, USA) at 4 °C for 12 h. After they had been washed three times with PBS, the PSCs were incubated with CY3-labeled (red fluorescence) goat anti-mouse IgG antibodies (ABconal, Wuhan, China) at 37 °C for 1 h. The cells were kept from light, stained with DAPI (blue fluorescence) for 10 min, and washed with PBS three times. Images were captured using a Leica SP8 microscope.

### 4.8. RNA FISH

A lncRNA FISH kit (RiboBio, Guangzhou, China) was used for an RNA FISH analysis of PSCs in accordance with the operating procedures. Images were captured using a confocal laser scanning microscope. The lncRNA *TCONS_00323213* FISH probe was synthesized by RiboBio, a biotechnology company(RiboBio, Guangzhou, China).

### 4.9. RNA Pull-Down and RIP Assays

For the RNA pull-down assay, T7 RNA polymerase (Roche, Mannheim, Germany) and biotin RNA labeling mix (Roche, Mannheim, Germany) were used to synthesize the five *TCONS_00323213* truncated fragments. A total of 2 µg of in vitro biotinylated RNAs were incubated with PSC protein lysis solution for 16 h. Then, the complex was pulled down using Dynabeads^TM^ M-280 Streptavidin (Invitrogen, New York, NY, USA). The beads were washed at least three times. Subsequently, the lncRNA–protein complexes associated with streptavidin magnetic beads were subjected to MS and western blot.

RIP assays were performed using an EZ-Magna RIP kit (Millipore, Billerica, MA, USA) following the operating procedures. Approximately 10^8^ PSCs were collected at 36 h of differentiation; cells were lysed with RIP lysis buffer supplemented with PMSF (Beyotime, Shanghai, China), and the lysates were incubated with 10 µg of the antibody to PKNOX2 (Proteintech, Wuhan, China) or IgG (ABconal, Wuhan, China; control) at 4 °C for at least 16–20 h. Then, Dynabeads^TM^ protein G (Invitrogen, New York, NY, USA) beads were added to capture the protein–RNA complexes.

### 4.10. CUT and Tag and RNA-Seq Analysis

A CUT and Tag Assay Kit for Illumina (Vazyme, Nanjing, China) was used to construct a CUT and Tag library for PKNOX2 in PSCs in accordance with the operating instructions, followed by Illumina sequencing. CUT and Tag analyses were performed according to the transcription factor processing chip-seq (https://github.com/nf-core/chipseq/releases) (accessed on 22 April 2021). The pig reference genome sequence file that we used was Sus_scrofa 11.1 (https://ftp.ensembl.org/pub/release-108/fasta/sus_scrofa/dna/Sus_scrofa.Sscrofa11.1.dna.toplevel.fa.gz) (accessed on 22 April 2021), and the genomic annotation file was downloaded from Ensembl (https://ftp.ensembl.org/pub/release108/gtf/sus_scrofa/us_scrofa.Sscrofa11.1.108.chr.gtf.gz) (accessed on 22 April 2021). The R package ChIPseeker [47] was used to identify the nearest genes around the peak and annotate genomic regions of peaks. All alignment results were then converted to coverage bigwig files and normalized to the corresponding input using deepTools [48]. The bigwig formats were visualized using the Integrative Genomics Viewer (IGV) [49] software. Motif analysis was performed using the Hommer software (v4.11) [50]. We analyzed RNA-seq of PSCs for 36 h of differentiation after the knockdown of *TCONS_00323213* (accession number: SRP186451). The reference genome and reference genome annotation file were similar to those used in the CUT and Tag analysis. RNA-seq read mapping efficiency is supplied in Appendix A. DEseq2 [51] was used to identify differentially expressed genes. Annotated genes showing |*log*2*FoldChange*|(|*log*2*FC*|) ≥ 1 and a *p*-value < 0.05 were considered to be differentially expressed.

### 4.11. Dual-Luciferase Reporter Assay

When cell confluence reached approximately 80%, the wild-type and mutant dual-luciferase reporter vectors of the *MyoG* promoter and PZW1, PZW1-*PKNOX2*, and PZW1-*TCONS_00323213* were separately co-transfected into C2C12 cells with the TK normalizing reporter plasmid. After incubation for 36 h, the firefly and renilla luciferase activities were measured using the Dual-Luciferase^®^ Reporter Assay System (Promega, Fitchburg, WI, USA).

### 4.12. Statistical Analysis

The results are presented as the mean ± standard deviation (SD). A one-way analysis of variance with post hoc Student–Newman–Keuls tests for multigroup comparisons of means was carried out. Two-tailed Student’s t-tests were performed using SPSS. Values of *p* < 0.05 indicated significant differences.

## 5. Conclusions

In summary, TCONS_00323213 is a vital regulator that affects proliferation and differentiation in PSCs. We present a molecular model to elucidate the function of TCONS_00323213 in regulating PSC differentiation through interactions with tran-scription factor PKNOX2, while relieves its inhibitory effect on the MyoG promoter. Our research provides new insights into the molecular mechanisms of lncRNA in porcine myogenesis.

## Figures and Tables

**Figure 1 ijms-24-06773-f001:**
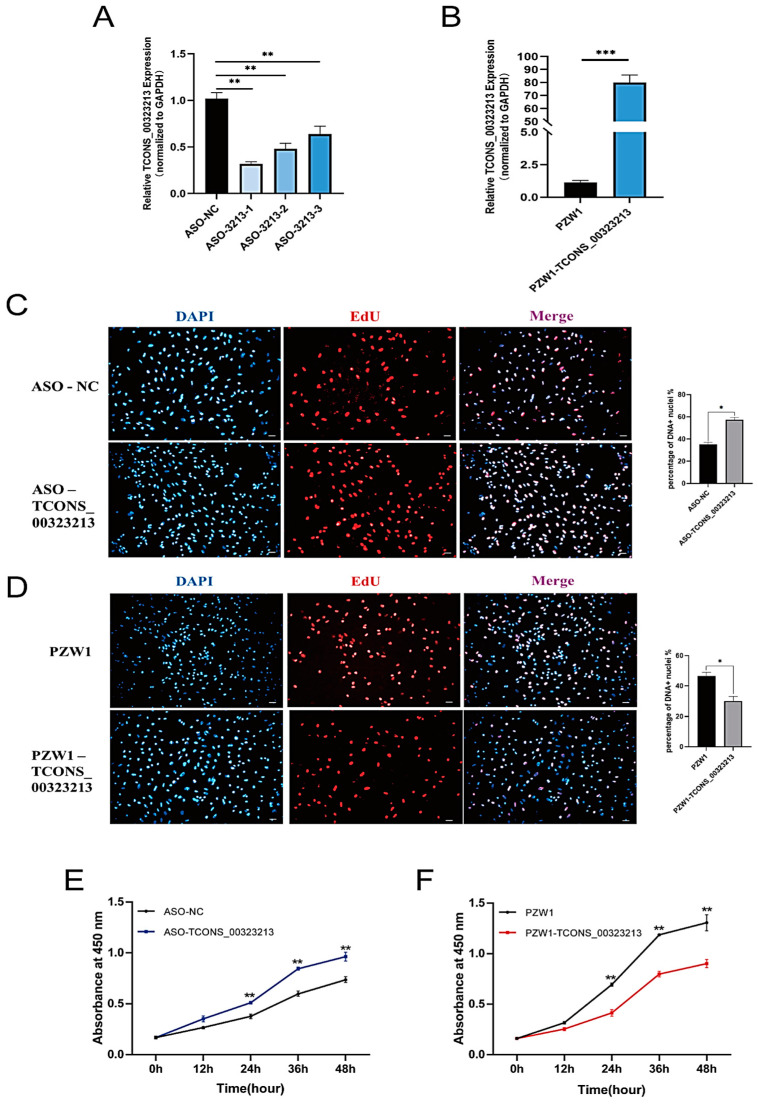
LncRNA *TCONS_00323213* inhibits myoblast proliferation. (**A**) *TCONS_00323213* knockdown efficiency detection. (**B**) *TCONS_00323213* overexpression efficiency detection. The PZW1 plasmid was used to construct the overexpression vector. (**C**) EdU staining assays after *TCONS_00323213* knockdown. (**D**) EdU staining assays after *TCONS_00323213* overexpression. Scale bar: 50 µm. (**E**,**F**) CCK-8 assays of PSCs suggested that *TCONS_00323213* knockdown significantly promoted myoblast proliferation after transfection with ASO-*3213*-1 for 12, 24, 36, and 48 h compared with proliferation in the NC group (**E**), while *TCONS_00323213* overexpression inhibited myoblast proliferation (**F**). Mean values ± SD, n = 3. Statistical significance was assessed by Student’s *t*-test. * *p* < 0.05, ** *p* < 0.01, and *** *p* < 0.001.

**Figure 2 ijms-24-06773-f002:**
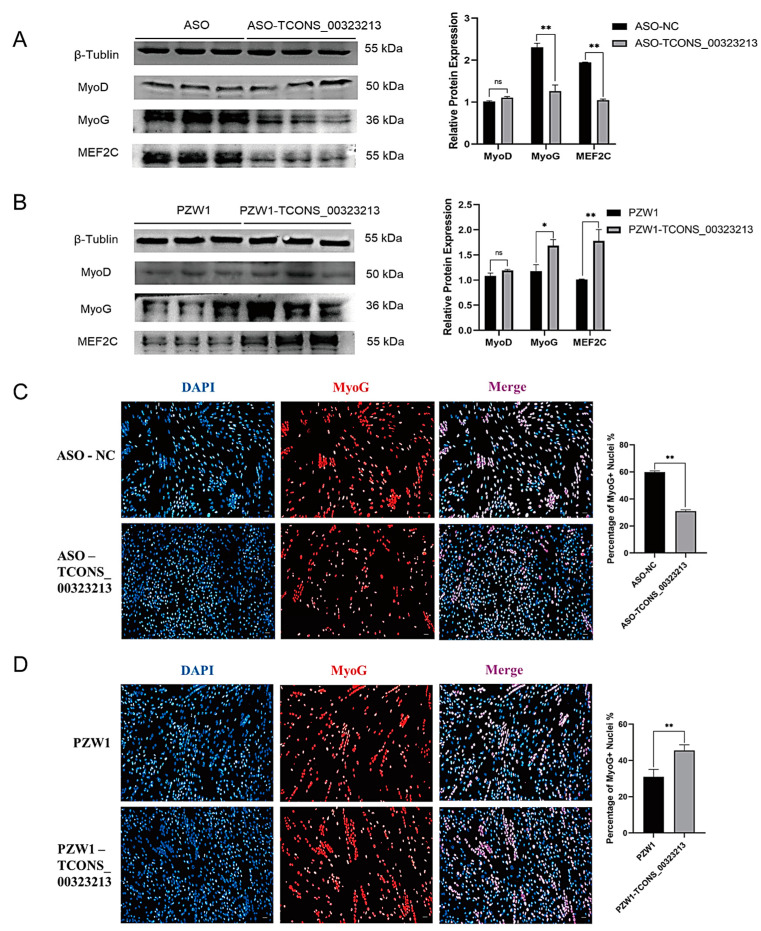
*TCONS_00323213* promotes myoblast differentiation. (**A**) Western blot analysis of MyoD, MyoG, and MEF2C expression levels after *TCONS_00323213* knockdown. (**B**) Western blot analysis of MyoD, MyoG, and MEF2C expression levels after *TCONS_00323213* overexpression. The western blot results contained three biological replicates in each group. (**C**) In PSCs differentiated for 36 h, a knowckdown of *TCONS_00323213* significantly decreased MyoG expression. (**D**) Immunofluorescence staining in PSCs differentiated for 36 h showed that *TCONS_00323213* overexpression significantly increased the MyoG expression level. Mean values ± SD, n = 3. * *p* < 0.05, ** *p* < 0.01. ns indicates no significant difference.

**Figure 3 ijms-24-06773-f003:**
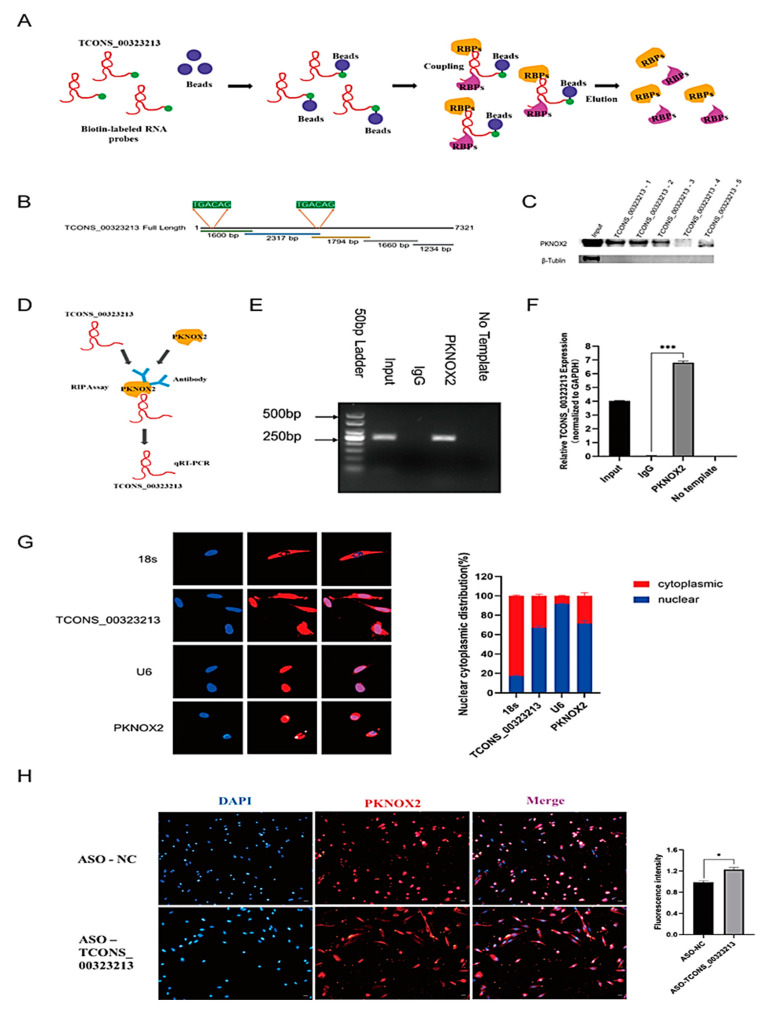
*TCONS_00323213* physically interacts with PKNOX2. (**A**) A schematic representation of the RNA pull-down assays. (**B**) Location of the *TCONS_00323213*-3 mutant fragments. (**C**) Interactions between a series of *TCONS_00323213* mutant fragments (*TCONS_00323213*-1, *TCONS_00323213*-2, *TCONS_00323213*-3, *TCONS_00323213*-4, and *TCONS_00323213*-5) were assessed by RNA pull-down and WB assays. (**D**) Schematic representation of the RIP assay. (**E**) RIP assays were performed to validate the interaction between *TCONS_00323213* and PKNOX2. (**F**) qPCR analysis of RIP assay results shows that *TCONS_00323213*’s relative expression is normalized to *GAPDH*. (**G**) Localization detection of *TCONS_00323213* and PKNOX2 in PSCs. The 18s was used as an internal reference for cytoplasmic localization, while U6 was used as an internal reference for nuclear localization. (**H**) Localization analysis of PKNOX2 expression in PSCs after knockdown of *TCONS_00323213*. Mean values ± SD, n = 3. * *p* < 0.05, *** *p* < 0.001.

**Figure 4 ijms-24-06773-f004:**
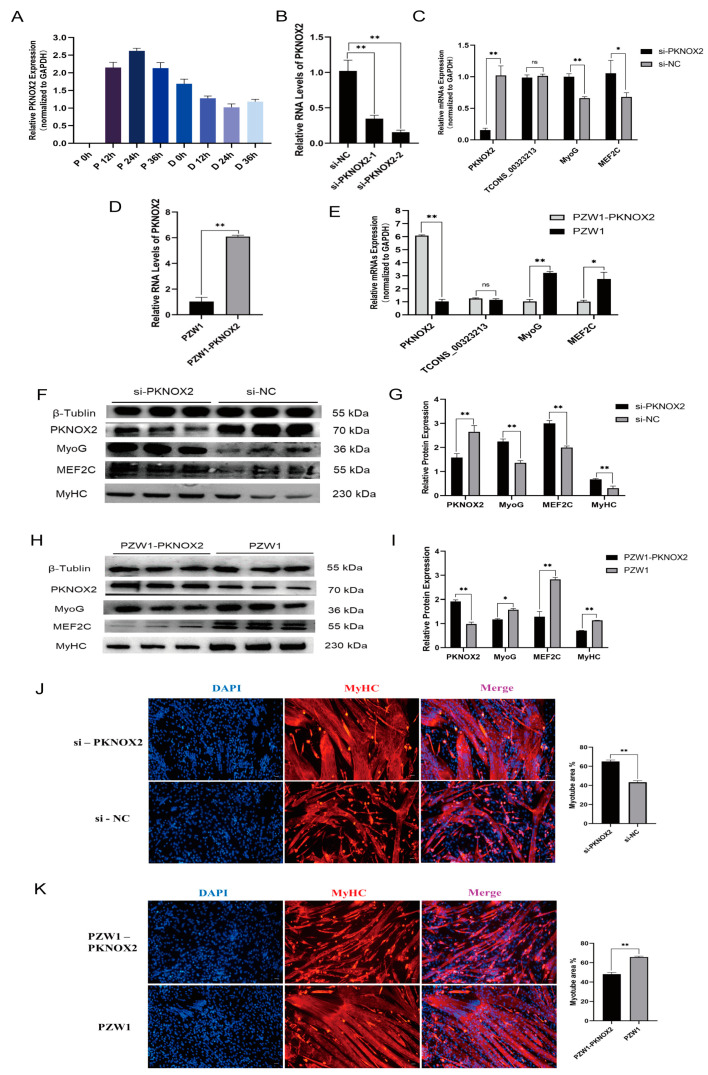
Role of PKNOX2 in the differentiation of PSCs. (**A**) Real-time PCR analysis of *PKNOX2* expression in PSCs during the period of proliferation and differentiation. There was a 12 h interval between each period. (**B**) A screening assay of siRNAs targeting *PKNOX2* showed that si-*PKNOX2-2* had the highest interference efficiency. (**C**) The knockdown of PKNOX2 increased the mRNA levels of *MyoG* and *MEF2C*. (**D**) Overexpression vector effect testing. (**E**) Overexpression of *PKNOX2* decreased the mRNA levels of *MyoG* and *MEF2C*. (**F**,**G**) Knockdown of *PKNOX2* increased the protein levels of *MyoG, MEF2C*, and *MyHC*. (**H**,**I**) Overexpression of *PKNOX2* decreased MyoG, MEF2C, and MyHC protein expression levels. (**J**) IF in PSCs differentiated for 36 h, showing that the MyHC expression level was significantly increased by PKNOX2 knockdown. (**K**) IF in PSCs differentiated for 36 h showing that PKNOX2 overexpression significantly decreased the MyHC expression level. Mean values ± SD, n = 3. * *p* < 0.05, ** *p* < 0.01. ns indicates no significant difference.

**Figure 5 ijms-24-06773-f005:**
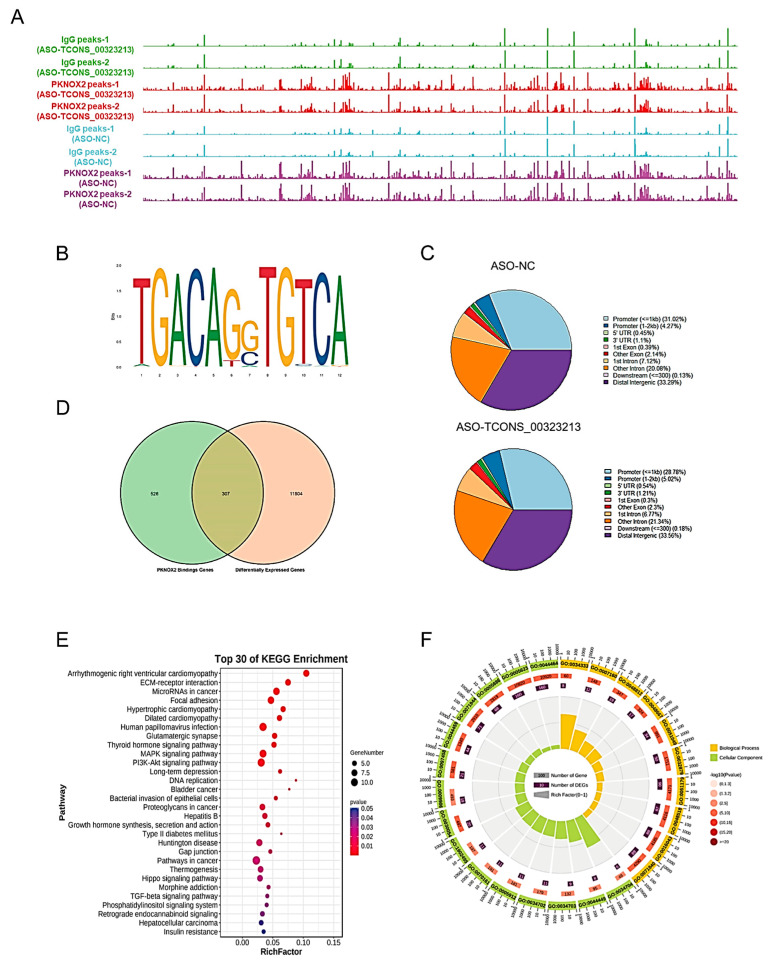
CUT and Tag Analysis of PKNOX2. (**A**) PKNOX2 enrichment analysis. (**B**) PKNOX2 significant enrichment motif sequence. (**C**). Distribution of PKNOX2-enriched regions on the genome. (**D**) Venn diagram of PKNOX2-binding genes versus differentially expressed genes in PSCs with *TCONS_00323213* knockdown. (**E**). KEGG pathway enrichment analysis of the 307 genes in panel (**D**). (**F**). GO enrichment analysis of the 307 genes in panel (**D**). The RNA-seq experiment had three biological replicates, and the CUT and Tag experiment had two biological replicates.

**Figure 6 ijms-24-06773-f006:**
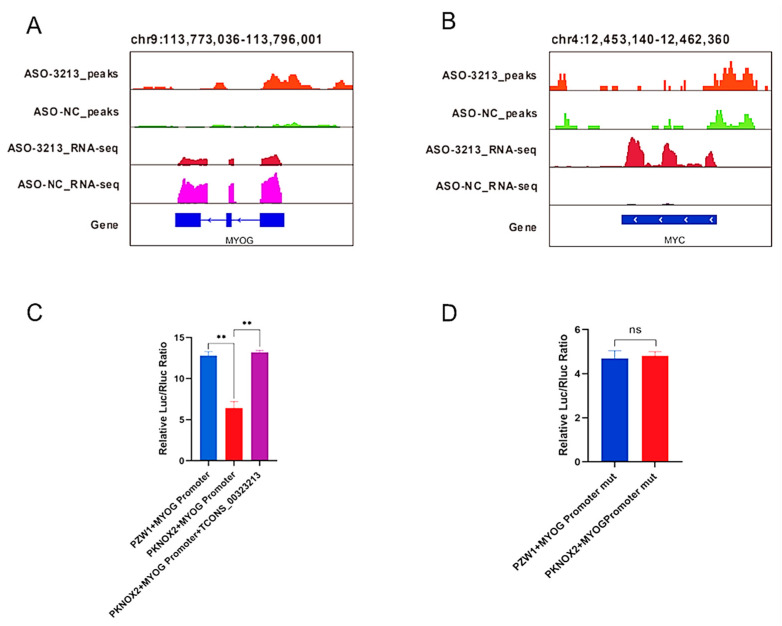
IGV visualization demonstrates the RNA-seq and CUT and Tag signatures of myogenic differentiation-related genes. (**A**) Enrichment signal of PKNOX2 on *MyoG* and differential expression of *MyoG* by RNA-Seq. (**B**) The enrichment signal of PKNOX2 on *MYC* and differential expression of *MYC* by RNA-Seq. (**C**) PKNOX2 overexpression reduced the luciferase activity of the wild-type *MyoG* promoter construct, and luciferase activity increased upon *TCONS_00323213* overexpression after PKNOX2 and the *MyoG* promoter plasmid had been co-transfected with the *MyoG* promoter. (**D**) Overexpression of PKNOX2 does not affect the luciferase activity of the *MyoG* promoter mutant. Mean values ± SD, n = 3. ** *p* < 0.01. ns indicates no significant difference.

**Figure 7 ijms-24-06773-f007:**
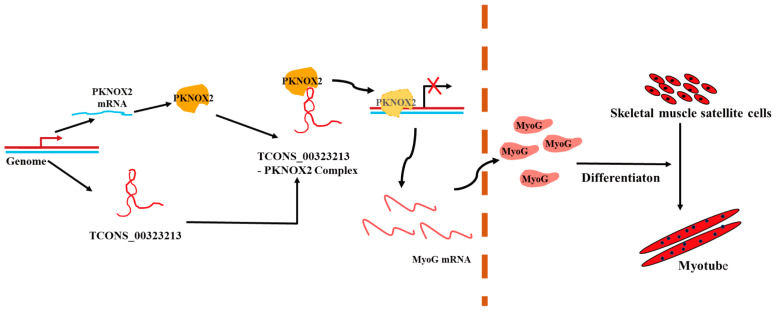
A schematic diagram depicting the functions of *TCONS_00323213* during skeletal muscle satellite cell differentiation. In differentiating PSCs, *TCONS_00323213* binds to PKNOX2 to relieve the inhibitory effect of PKNOX2 on *MyoG*, thereby increasing *MyoG* expression and promoting PSC differentiation.

## Data Availability

The data presented in this study are available upon request from the corresponding authors.

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
