# Peer review of "LncRNA TCONS_00323213 Promotes Myogenic Differentiation by Interacting with PKNOX2 to Upregulate MyoG in Porcine Satellite Cells"

_ijms, 2023, doi:10.3390/ijms24076773_

Round 1

Reviewer 1 Report

ijms-2214685

Title: LncRNA TCONS_00323213 Promotes Myogenic Differentiation by Interacting with PKNOX2 to Up-regulate MYOG in Porcine Satellite Cells

Authors: Mengxun Li, Quan Liu, Su Xie, Chog Fu, Jiaxuan Li, Cheng Tian, Xin Li, Changchun Li *

In general, the data in present studies are good and support the major conclusions of this manuscript. However, following issues need to be considered prior to considering the manuscript of publication.

[Major concerns]

1. Abbreviations: The use of abbreviations when writing a paper has many advantages besides simplicity of expression. To use an abbreviation, first write the abbreviation in parentheses after the full name, and then use the abbreviation from Introduction to the final Conclusion. Only in Abstract and Figure legend do it separately. In particular, because of the characteristics of IJMS, where Materials and Methods is arranged at the end of the paper, the original words and abbreviations are written in the order they are used from the introduction, and only when the abbreviation is used repeatedly, the abbreviation can be used until the conclusion.

In the abstract, MYHC and MEF2C do not need additional abbreviations, so they are not necessary. In other words, an abbreviation is necessary for repeated use, whether in the abstract or in the main body of manuscript.

And since so many abbreviations are used in this article, please reorganize them in full name (abbreviation) form very thoroughly from the introduction.

2. In the method of labeling human body proteins and genes, genes should be italicized differently from proteins, but this is not followed at all, leading to confusion, so please correct all of them.

3. Statistical analysis method: As far as I know, the usual statistical comparison used here is that p value < 0.001 is appropriate. * p < 0.05, ** p < 0.01, and *** p < 0.001. If you have any questions about this, ask a statistician around you.

4. English: There are many typos. There are also many cases where the first letter of a word is capitalized even though it is not a proper noun.

5. In the method of labeling human body proteins and genes, genes should be italicized differently from proteins, but this is not followed at all, leading to confusion, so please correct all of them. Precise English proofreading by an expert who knows this field well is needed.

6. Materials and Methods section - When naming a particular chemical company, you must provide location information such as company name, city and/or state (abbreviation in the USA and Canada) and country. Once you have named a company with the information, you should only mention a company’s name thereafter. Information about several companies is wrong, so check and correct it. It is generally well written in this paper, but there are a few mistakes, so find them and correct them. Examples: Millipore, etc.

[Minor concerns]

1.    Line 50: Myofibroblasts should be written as myoblasts.

2.    Line 54: Define IGF2.

3.    Line 55: Define PSC here again.

4.    Line 56: Define MyoD.

5.    Line 61: Define SRF, lncSYISL, and lncMGPF. MyOD should be written as MyoD and re-write this sentence.

6.    Line 70: Define E2F3.

7.    Line 75: Define TTN.

8.    Line 76: 24h should be written as 24 h.

9.    Lines 89, 97, and 98: EDU should be written as EdU.

10.    Line 107: The sentences written by the authors seem to be citing prior results. However, since you are actually referring to Figure S2, that is, supplementary data, rewrite this sentence.

11.    Line 111: ‘TCONS_ 0032321’ should be ‘TCONS_00323213’ without a space.

12.    Figure 2A and 2B: Always define ‘ns’ in the figure legend whenever it appears in the figures.

13.    Figure 3F: The shape of Figure 3F is not normal, please correct it again.

14.    Line 175: ‘thePKNOX2’ should be written as ‘the PKNOX2’.

15.    Line 239: ‘CUT&Tag’ should be written as ‘CUT & Tag’.

16.    Line 316: Re-write the sentence.

17.    Line 317: Time unit such as ‘hours’ and ‘h’ are used. Try to use just one form.

18.    Line 326: Re-write the unit of temperature with Time New Romans. Similar mistakes are everywhere.

19.    Line 326: CO2 should be CO2.

20.    Line 397: ‘deviation(SD)’ should be ‘deviation (SD)’.

21.    Reference section: Author should consult and peruse carefully recent issues of the journal, International Journal of Molecular Sciences (IJMS), for format and style. Also double-check the abbreviations of journal names. There are even many missing pages in several references.

22.    In the case of mentioning the Epub date in the reference, it indicates the reference in the case of Epub before the actual publication of the paper. In other words, if the paper was formally published, the Epub date is meaningless, so delete them all. And there is a notation [J] at the end of the title of several referenced papers, but I don't know what it means, but delete them all.

Overall, the manuscript can be considered to publication after major revision as indicated above.

Reviewer 2 Report

The current manuscript by Li M et al. describes the role of lncRNA TCONS_00323213 during myogenic differentiation of porcine satellite cells. It has been shown that TCONS_00323213 interact with PKNOX2 homeobox protein to regulate expression of critical genes involved in myogenic differentiation. Though the current findings are interesting, the manuscript suffers with missing of methodological details, inconsistencies in results provided, and poor writing. The proposed mechanism needs some more experiments to validate it. Overall, the manuscript is carelessly written. Reviewer enthusiasm limited due to the following concerns:

Introduction:

1)     Introduction for lncRNA TCONS_00323213 (such as genomic location; homology; etc.)  needs to be added in addition to general intro of various other lncRNAs.

2)     In lines 73-75, authors stated that upregulation of TCONS_323213 during myogenic differentiation reported previously. But, failed to cite the previously published research.

3)     If the results provided in Figure S1 and S2 are reported/published earlier, no need to add the results again. Remove the previously published data from supplementary and just cite the original research.

4)     What is ‘TTN’ in line-75. Is it gene or abbreviation for something. Please elaborate.

5)     ‘TTN’ in line-75 and ‘MYHC’ in line-76 were not found in pig genome assembly. Add the correct official name.

6)     Please cite and discuss the recent paper by Liu J et al. (PMID: 36253723) that described the TCONS_323213 lncRNA role in myogenic differentiation by regulating gene expression and alternative splicing.

Results:

7)     What is ‘PZW1’ indicated in Figure-1? Add the details in the figure-1 legend and corresponding results described in section 2.1.

8)     Check the Figure 1B bar graph y-axis legend. It indicates as %, but denotes relative values ranging from 0-0.6. Correct it to % similar to that of Fig-1A.

9)     Please add the qPCR plot to show the efficiency of TCONS_323213 knockdown and or Overexpression after transfection.

10)  Please add the n=? details for EdU assay quantifications showed in Figure-1 in the legend.

11)  In line 108, Correct the ‘Figure S2’ to ‘Figure-S1’.

12)  In line 108, authors stated as ‘RNA-Seq’, but the Figure-S2 describes only qPCR. Correct the statement. If this data published earlier by the authors, please cite it instead of publishing it again.

13)  Figure-2A and 2B results indicate ‘triplicates’ or ‘three’ different clones’? Add the details in the legend.

14)  Please add the n=? details for MYOG IF assay quantifications showed in Figure-2 in the legend.

15)  Add the colocalization quantification-in Figure 3A to in addition to just to show merged image of one layer.

16)  It is not clear whether knockdown of TCONS_323213 or PKNOX2 affect their localization to the nucleus. Supplementing that results to the Figure-3 would highlight the role for their interaction and function.

17)  It is not clear how did authors ended up checking PKNOX2 directly for the FISH studies? What is the lead for selecting PKNOX2? Start section 2.3 with the lead experiment.

18)  Figure-3G y-axis legend indicates ‘relative expression’. Relative to what? Indicate clearly in the legend.

19)  Is it PKNOX2 is human protein used for RIP? What is the name of PKNOX2 pig ortholog gene name? PKNOX2 is not listed in pig genome assembly.

20)  Supplementing Figure-4 with the qPCR expression profile of PKNOX2 during myogenic differentiation (similar to Figure-S1B) would indicate the rationale for studying PKNOX2 role in differentiation after knockdown.

21)  Also, add the qPCR levels of TCON_323213 lncRNA after PKNOX2 knockdown during differentiation to reveal its transcriptional dependency of PKNOX2-TCONS_323213 (in Figure-4)

22)  In line 206, check ‘Figure 5v’ and correct it.

23)  Relabel the Figure-5A tracks instead of default file name of bigwig files. It is hard to read the names.

24)  What does the Figure-5B motif denotes? Is it specific for any transcriptional factor or PKNOX2 itself? Only one motif is enriched throughout or authors had selected it based on some cut-off? Add the details.

25)  In Figure-6, check PAX7 gene. Is it PAX6? Pig genome assembly only shows PAX6. Check the name and correct it.

26)  Figure-6A-D shows some RNA=Seq tracks after TCONS_323213 K, but the corresponding methodological details were missing. What day after differentiation and TCONC ASO KD, the RNA-Seq performed?

Discussion:

27)  Discussion needs to improved with relevant literature. Cutdown some repetition of Intro and results in the discussion.

28)  discuss the recent paper by Liu J et al. (PMID: 36253723) that described the TCONS_323213 lncRNA role in myogenic differentiation by regulating gene expression and alternative splicing. Correlate the DEGs authors obtained with that of the study published by Liu et al.

Materials and Methods:

29)  Add ‘methodological’ details how did authors performed transfection on PSCs. How did the transfection efficiency was ensured and normalized between ASO-NC and ASO-TCONS & PZW1 & PZW1-TCONS/ PKNOX2 siRNAs?

30)  Authors have used ‘Luciferase-reporter’ assays in Figure-6, but the methods are missing. Please add the details.

31)  In section 4.9, authors indicated RNA pull-down assays. But the corresponding results were not provided or described in the manuscript.

32)  In line 389, check ‘108 cells’? is it 10e8 (100 million cells)?

33)  Also, in line-389, authors indicated 10ug of PKNOX2 antibody (which is specifically detects human/rat/mouse version). Did authors check TCONS_323213 interaction with endogenous PKNOX2 or recombinant purified protein? If so, add the source of recombinant pure protein that is used for RIP assays.

34)  Add more details for different packages/software’s used for CUT and Tag data analysis in section 4.10.

35)  Indicate which version of pig genome assembly used for visualizing RNA-Seq and CUT &Tag assay tracks in figure-5 and figure-6.

36)  RNA-Seq results were described in Figure-6, but the corresponding methods were missing. Experimental details and bioinformatic analysis/pipeline used for the analysis needs to be provided.

Other comments:

37)  Check the Figure S1D bar graph y-axis legend. It indicates as %, but denotes relative values ranging from 0-1. Correct it to %. Also add the ‘color legend’ to denote which color denotes nuclear /cytoplasmic fraction.

38)  Same thing with Figure S2, ADD the ‘color legend’ to denote which color denotes control and ASO clearly.

39)  Similar to the original images for agarose gel (Figure-3F), provide the original blots for different westerns showed in Figure-2 and Figure-4 in the supporting information.

40)  Table S1 and S2 needs to be renamed in the order of their first appearance in the text. Table S2 appear first at line-176 and Table S1 comes later at line-352.

41)   Supporting information submitted along with the paper, i.e. Table-S3/ Table-S4/ and Table-S5 were not cited in the manuscript. Check carefully and cite them in the manuscript.

42)  Manuscript NEEDS to be checked for grammatical accuracy and language corrections throughout.

Best wishes,

Round 2

Reviewer 1 Report

Accept in present form.

Reviewer 2 Report

Dear authors,

Thanks for providing the revised version of your manuscript and addressing most of reviewer concerns. 

In lines 82-87, authors stated that up-regulation of TCONS_323213 during myogenic differentiation reported previously. But, failed to cite the previously published research. 

Please check the manuscript for textual accuracy and figures/tables citation in the revised version.
